

# A deep learning based approach for extracting Arabic handwriting: applied calligraphy and old cursive

Saber Zerdoumi[1,2], NZ Jhanjhi[2], Riyaz Ahamed Ariyaluran Habeeb[3] and Ibrahim Abaker Targio Hashem[4]

[1] Research Unite Cerist, Université Constantine, Constantine, Algeria
[2] School of Computer Science, SCS, Taylor's University, Subang Jaya, Malaysia
[3] Department of Computer Science and Information Technology, Taylor's University, Kuala Lumpur, Malaysia
[4] Department of Computer Science, The University of Sharjah, The Sharjah, UAE

Corresponding authors
NZ Jhanjhi, Noorzaman.jhanjhi@taylors.edu.my
Ibrahim Abaker Targio Hashem, ihashem@sharjah.ac.ae

## ABSTRACT

Based on the results of this research, a new method for separating Arabic offline text is presented. This method finds the core splitter between the "Middle" and "Lower" zones by looking for sharp character degeneration in those zones. With the exception of script localization and the essential feature of determining which direction a starting point is pointing, the baseline also functions as a delimiter for horizontal projections. Despite the fact that the bottom half of the characteristics is utilized to differentiate the modifiers in zones, the top half of the characteristics is not. This method works best when the baseline is able to divide features into the bottom zone and the middle zone in a complex pattern where it is hard to find the alphabet, like in ancient scripts. Furthermore, this technique performed well when it came to distinguishing Arabic text, including calligraphy. With the zoning system, the aim is to decrease the number of different element classes that are associated with the total number of alphabets used in Arabic cursive writing. The components are identified using the pixel value origin and center reign (CR) technique, which is combined with letter morphology to achieve complete word-level identification. Using the upper baseline and lower baseline together, this proposed technique produces a consistent Arabic pattern, which is intended to improve identification rates by increasing the number of matches. For Mediterranean keywords (cities in Algeria and Tunisia), the suggested approach makes use of indicators that the correctness of the Othmani and Arabic scripts is greater than 98.14 percent and 90.16 percent, respectively, based on 84 and 117 verses. As a consequence of the auditing method and the assessment section's structure and software, the major problems were identified, with a few of them being specifically highlighted.

## INTRODUCTION

The increased use of images as instruments for distributing information over the past two decades has necessitated the requirement for image-type data acquired by different devices to be managed and interpreted. Furthermore, the rapid expansion of side effect of online media as a whole. in an avalanche of structured and unstructured data (such as images).

This study proposes a unique technique for segmenting Arabic offline text that takes into account the core separation of the "middle" sections from the "lower" zone by recognizing a character's strong degeneration in zones. The baseline, as a key, serves as a delimiter for horizontal projections in addition to script localization, including the essential functionality that detects the direction of the beginning point. To differentiate modifiers in zones among the well-known approaches, optical character recognition (OCR) (*Srihari, Shekhawat & Lam, 2003*) is regarded as an acknowledged (as well as the standard) method for grasping and analyzing many languages with varying features and difficulties, such as Chinese (*Du & Huo, 2013*; *Patil & Shimpi, 2011*), English (*Bag, Harit & Bhowmick, 2014*; *Saber et al., 2017*). Text recognition systems with high recognition rates have been created by researchers. Character-segmentation-required Arabic handwriting is provided. For the Arabic recognition task, it is your responsibility to extract Arabic text from photos. These varying styles of writing were, on the other hand, retained as intricate patterns and materialized as visuals or films (*Saber et al., 2017*). Handwriting recognition technologies are often used to decipher written text from a picture. In the early days, OCR techniques were utilized, and the results were passed on to specialized engines to search for specific terms. In spite of its promising beginnings, the research has encountered a number of roadblocks. Classifiers such as support vector machines (SVMs), HMM, recurrent neural networks (RNNs), and neural networks (NNs) have been used to tackle handwritten Arabic word recognition (*Bataineh, Abdullah & Omar, 2011*; *Hakak et al., 2019*; *Aouadi, Amiri & Echi, 2013*). To differentiate handwritten Arabic words *Parvez & Mahmoud (2013)* and *Mahmoud et al. (2014)* discovered syntactic and structural characteristics. Text line slant angles were manipulated as part of their research. In addition, they suggested introducing a "new design segmentation technique" into the recognition process (*Hakak et al., 2019*; *Aouadi, Amiri & Echi, 2013*; *Supriana & Nasution, 2013*; *Chherawala & Cheriet, 2014*) emphasize the recognition of cursive handwriting using several approaches that use either a shape-based or a descriptor-based approach. A mix of genetic propagation and artificial NN-based approaches enhances cursive handwriting recognition accuracy (*Garg & Bajaj, 2015*). The goal of this study is to propose new approaches for synthesizing learning approaches for recognizing Arabic word features *via* zone segmentation, as well as to provide a comprehensive approach that addresses the challenges of extracting Arabic cursive writing, including calligraphy recognition systems. The research also looks at the various approaches used throughout the segmentation and classification stages. The effectiveness of current systems is assessed. There is a discussion of issues with Arabic cursive writing recognition systems. Most notably, the many processes required to develop a generic Arabic writing recognition system are described. The remainder of this work is structured as follows. 'Related work' provides context for Arabic cursive writing recognition challenges, including associated terms, definitions, and Arabic language features. 'Segmentation' presents a zone-based method to word recognition. 'Feature extraction' presents an overview of cutting-edge feature extraction approaches. The recognition step of the suggested paradigm for Arabic character recognition is presented in 'Proposed approaches for zone-based word recognition'. 'Feature extraction' closes the report by outlining possible future research objectives in this field.

## RELATED WORK

This section reviews and summarizes several current developments on image processing. These are Arabic character segmentation and recognition methods.Sub-methods in segmentation are only concerned with pattern recognition in images. Overlapping, close-character issues, composite and touching characters are some of the approaches tried to Arabic handwriting. The initial phases in each of these procedures, These three processes deal with Arabic text lines: digitization, noise reduction, and skew detection. Processing corrections are necessary. Segmentation is necessary to recognize text in an image; the whole text is translated into lines of text, and subsequently into single words. Vertical and horizontal projections are used (*Mozaffari et al., 2008*) to break up long strings of text. In spite of these difficulties, research in this area is only getting started. Three factors make constructing Arabic OCR systems difficult. To begin, Arabic cursive writing is printed from right to left (*Patil & Shimpi, 2011*). Second, the shapes of Arabic letters vary depending on whether the alphabet is connected from the beginning to the middle or from the middle to the end. Finally, Arabic scripts are represented by a variety of fonts in printing machines and recognition systems. A variety of factors lead to the necessity for a system specifically specialized to the recognition of Arabic writing. Orthography and pronunciation of Arabic characters.

The Arabic language is similar to Latin and has alphabets named Hourouf (حروف) (possesses 28 characters). It does, however, have its own morphological form, language orthography, and character morphology. Each figure has two to four unique forms, which are selected at random.

It is based on linking characters inside few words\sub-words. Forms are used to represent positions. The single alphabets may be swapped from beginning to middle, halfway through (sub-)words, or by isolating (sub-)words. Table 1 depicts each design. By rewarding surrounding characters on each side and linking them in their neighborhood, the shape of each raw character may be transformed depending on its surrounds based on its sides. Several characters' initials or middle names change. They are simply appended to the end of the (sub-)words.

It does, however, have its own morphological form, character morphology and linguistic encoding. Each figure has between two and four distinct shapes that are randomly generated. Linking words and sub-words together is the foundation of this technique. Forms are used to represent positions. The single alphabets may be swapped from beginning to middle, halfway through (sub-)words, or by isolating (sub-)words. Table 1 depicts each design. By rewarding surrounding characters on each side and linking them in their neighborhood, the shape of each raw character may be transformed dependent on its surrounds and sides. Several characters' initials or middle names change. They are simply appended to the end of the (sub-)words.

The retrieved attributes serve as the foundation for the Arabic cursive identification method. To offer a clear picture of each sequential step in recognizing features from the input source to the end stage, which includes per-processing, segmentation, classification, and presenting case studies on recognition rates by bench-marking with state-of-the-art

**Table 1  Arabic alphabet and its appearance in different positions (beginning, middle, and ending).**

| Isolated alphabets | | Right sound | Beginning (B) | Middle (M) | Ending (E) | Pronunciation when a character is combined with vowels (a, o, and i) |
|---|---|---|---|---|---|---|
| Alfeٱ | | A | ٱ | ـا | ا | Aa, Ao, Ai |
| Baa | ب | B | بـ | ـبـ | ـب | Ba, BO, BI |
| Taa | ت | T | تـ | ـتـ | ـت | TA, TO, TI |
| Thaa | ث | TH | ثـ | ـثـ | ـث | THA, THO, THI |

Arabic letter recognition systems, this debate on Arabic word recognition using image recognition systems is being conducted. This section is divided into two phases, the first of which is per-processing, which adds images into the character processing levels.

## Process flow of pattern recognition

The recognition process may be separated into stages that are stretched out across a series of steps. To grasp the features of each component, many sorts of recognition methods were examined. The explanation of the recognition stage is crucial since large-scale resources are needed and extracted during this step. There are five stages involved in pattern recognition: Prior to segmentation, I conduct per-segmentation. Segmentation, feature extraction, and post-segmentation are the following: Refer to Fig. 1 for a depiction of the four key recognition phases for Arabic recognition systems.

## SEGMENTATION

Segmentation as phase of the word recognition would not be possible without this mechanism. In the segmentation process, all attributes are divided into discrete pieces. It is broken down into units like pages, lines, words, and individual characters. This section discusses data rectification and data performance, as well as the difficulties and variety of segmentation strategies utilized in each subcategory. Document scanning or the collection and division into sub-paragraphs of single-page paragraphs is the starting point for segmentation in the case of machine-printed offline Arabic writing. A collection of sentences in a sub-paragraph is made up of string characters that are comparable to or the same as those in the sub-paragraph in issue (*Jayech, Mahjoub & Amara, 2016*). Noise reduction, document orientation detection, skew detection and deletion, and layout analysis are some of the per-processing-steps needed before segmentation can begin. Segmentation in Arabic writing created by computers has the structure of paragraphs separated into three levels: text lines, text lines of words, and single characters. Size and shape are important considerations when choosing a recognition technique (*Naz et al., 2016*) outlines a recognition method for printed Arabic texts. To get from 2D to 1D, we employ the vertical projection technique. In word characters, the assembly stroke is usually thinner than the other portions. This character segmentation technique is utilized in many different procedures to discover segmentation spots in sparsely traced texts (*Li et al., 2016*). A character's skeleton supplies vital information about the character's shape.

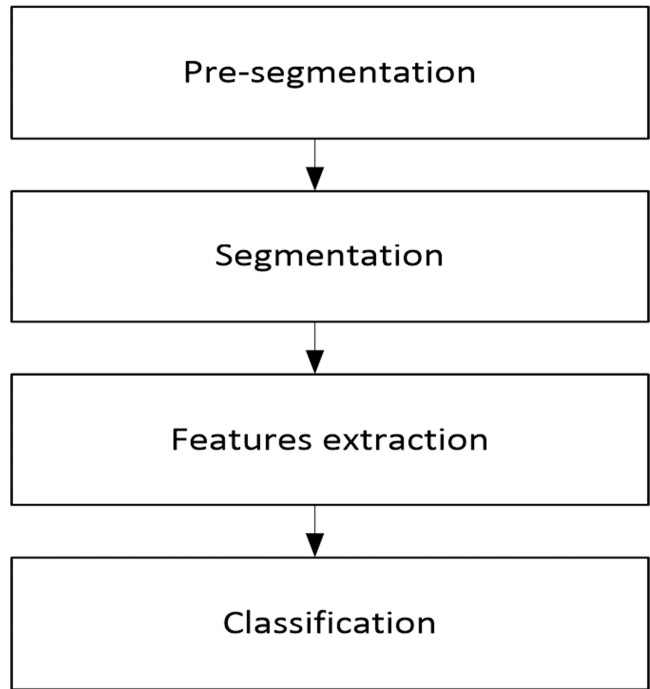

**Figure 1** **Diagram of the four main recognition steps for Arabic recognition systems.**

From earlier work (*Srihari, Shekhawat & Lam, 2003*), the 10 most frequent Arabic fonts were selected, and identification algorithms based on invented diaeresis segmentation and deluge punctuation marks. Clusters notation segmentation promises to be error-free and up to 98.73 percent accurate (*Vakil, Megherbi & Malas, 2016*). A technique known as K-means clustering (*Vakil, Megherbi & Malas, 2016*) is utilized as line divides are not very well divided or when instructions are on the outside of aggregate. The primary advantage of K-means is its mathematical efficiency, and it is a logical and confidence approach which has influenced many academics from other fields. The results of a K-means loop have validated. *Jin et al.*'s (*2005*) approach. On the other hand (*Gao et al., 2016*), the work was leading the research on unsupervised subspace learning via analysis directionally that was used in different clusters and calcification, which led to carrying out the experiment for a recognition system with a higher confidence rate. For recognizing printed Farsi texts, which also applies to printed Arabic text, "local baseline segmentation" for each sub-word is required. To put the suggested approaches to the test, a collection of printed Farsi text in over 20 distinct typefaces was used.For recognizing printed Farsi texts, which also applies to printed Arabic text, "local baseline segmentation" for each sub-word is required. To put the suggested approaches to the test, a collection of printed Farsi text in over 20 type There were no unique fonts utilized. The accuracy percentage for properly segmented related characters was 98.5 percent, while the accuracy rate for Farsi newspaper headlines was 100 percent. However, the method is font-dependent, with smaller font sizes resulting in poorer identification rates. An early effort (*Dai et al., 2016*) segmented Uygur characters,

a sub-Turkish language spoken in western China as Xingjian. The process was divided into two parts: topological segmentation and quasi-topological segmentation. Topological segmentation necessitates the tracing of outer boundaries (*i.e.,* contours of entire words produced from character edges and strokes of superior zones) prior to checking for any possible sideways interruptions for the vertical projection (*i.e.,* contours of whole words derived from character edges and strokes of superior zones). The quasi-topological technique combines character phases with feature extraction. The earliest version of the IRAC system created by *Amin, Al-Sadoun & Fischer (1996)*, *Tagougui, Kherallah & Alimi (2013)* was a parametric description based on topological and statistical properties retrieved from full words, which did not need segmentation. *Saabni, Asi & El-Sana, (2014)* recently adopted a holistic method instead of segmenting words into individual letters. A lot of filters are used hierarchically to minimize search space. A dynamic-time warping recognizer is used to features extracted from global geometries. Following that, the recognizer is used to choose and organize training set (alphabet) that fit the input data. Recognition of Arabic text is accomplished by the use of layer-based and block-based segmentation methods. To generate form masks, characterize appearance, arrange depths, and evaluate segmentation classes and instances, the layered approach detects the segmentation object and assembles detector outputs. Arabic handwriting segmentation is more challenging (*Jayech, Mahjoub & Amara, 2014*) because of the complexity of pattern processing and the time required to analyze databases. Following that, all isolated single characters are transformed into words, phrases, and number patterns. Due to Arab forms' geometrical restrictions, specializations in script and individual differences in handwriting hinder the process. When compared to other languages with known recognition rates, Arabic script has a similar recognition rate to that of English (*Shaikh, Mallah & Shaikh, 2009*). Given the difficulty with cursive Arabic text recognition, various approaches for handwritten Urdu and Farsi scripts were developed, primarily to identify similar script characters (*Pal, Jayadevan & Sharma, 2012*; *Saeed & Albakoor, 2009*). Writing, like other research (*Parvez & Mahmoud, 2013*), may correct a range of mistakes However, words and numerals should not have been split. Other methods for recognizing handwritten Arabian words include HMM classifiers and neural networks of support vector machines (*Dehghan et al., 2001*). Skew detection are a type of technology used to estimate the center of gravity (COG) of Arabic handwriting (*Atallah & Omar, 2008*) COGs are connected to the primary source and utilized to identify the skewed angles of scanned sheets by using the four extreme corner points of a text. Over 150 Arabic articles were extensively examined, yielding an overall accuracy skew percentage of 87%. For the nearest-neighbor classification technique, Arabic and Indian digits with HMM (*Mahmoud, 2008*) were employed. Single-digit items were employed to achieve a recognition rate of 97.99 percent HMM and 94.35 percent nearest-neighbor classifier, which is higher than most other classifiers. Following the completion of testing with the standard database, (*i.e.,* 10 HMM statistics), 44 writers developed 48 databases comprising 120 feature vectors. Each vector represents a digit (*i.e.,* a total of 21,120 digits). These sentences were taken directly from the article's text. The approach of partial projection rounds (*Mahmoud, 2008*) was used to segment unreliable Arabic handwriting. You'll need to draw a line along the contour at an angle in the setting direction and then run it in

the other direction using an Arabic handwriting database made up of 100 basic phrases. Another issue identified in handwritten Arabic lettering was the difficulty of touching and overlapping connected components (*Mahmoud, 2008*). Detecting unclear letters created by characters touching or overlapping is critical in Arabic handwriting because it may inhibit successful segmentation. A case study presented these strategies, which were established by integrating and overlapping relevant components in handwritten Arabic texts. For the technique to be effective, form identification and modification are also required for line segmentation. For simplified computations, edge points and visual forms impact character fitting, which may be simulated to provide a fixed variable N. In this method, a character's neighbor implies the connection among comparable patterns and the gathered selection of elements. Researchers presented a horizontal projection and profiling-based strategy for text line segmentation in *Al-Dmour & Fraij (2014)*, which leveraged autocorrelation to boost natural outline similarities. Although line–word extraction is dependent on the shape of Arabic handwriting, the technique outlined above assists in analyzing text line spacing. Based on training and testing in the AHDB database with this clustering technique, the extraction rate is 84.8 percent (*Al-Ma'adeed, Elliman & Higgins, 2002*).In order to deal with employment of ligature divide for shape and interconnections for slopes and curvatures in the basic segmentation stage allows Arabic cursive writing, as is similar to Urdu, to utilize a range of connected components (*Rabi, Amrouch & Mahani, 2018*). With a 98.75 percentile average accuracy, the "complex" function was used to eliminate important components (*Khorsheed, 2007*). After having to pass slope diagnosis in both the introductory and reverse direction and adopting a segmentation concept for unclear Arabic printed handwriting texts, lines were extracted from texts (*Bouchiareb, Bedda & Ouchetai, 2006*) using partial projection iterations and partial contours after passing slant detection in both the beginning and opposite directions. To separate the numbers, we used the Arabic handwriting database with 100 short words, as mentioned in the earlier portion of this research. The typical methods and issues of handwritten Arabic cursive writing, as well as the touching and overlapping connected components, were described (*Al-Dmour & Fraij, 2014*). Edge points and shapes have a direct impact on character matching, and they may be modeled to provide a fixed number N for easy computation. Similar shapes are created by the neighborhood and its gathered subset points. A text line segmentation technique based on horizontal projection and profiling (*Al-Dmour & Fraij, 2014*) was also proposed, which employs auto-correlation.

To increase the natural likeness of outline outlines and help in the evaluation of text line spacing. The nature of Arabic handwriting has an impact on line–word extraction. The clustering strategy proved to be the most effective for extracting data from the AHDB database. Because the form of ligature division and the connection of slopes and curvatures for fundamental segmentation are comparable in Urdu and Arabic cursive writing, numerous kinds of linked component approaches were applied (*Ali et al., 2004*). To remove related components, the compound feature was applied (*Rabi, Amrouch & Mahani, 2018*), and it had a 98.75 percent accuracy rate.

# FEATURE EXTRACTION

The stages involved in character feature extraction in offline and online handwriting and machine-printed recognition activities are referred to as "feature extraction" in the industry. The final stage before the recognition process is complete is known as "feature recognition". By definition, feature extraction is the process of extracting the material that the recognizer recognizes, such as pixels, form data, or mathematical qualities, which is sometimes applied for partitioning. Following the determination of feature extraction (*i.e.,* how good the features are), values are supplied to match letter samples for Arabic recognition or to assess if the characters fit to any matching classification at all in order to meet the separating goal for dissimilar objects. The sort of issues revealed by the sample text influence the features applied. Issues may arise with the processing system, which is critical in selecting features, whether offline or online, as well as with the cursive forms of handwritten or printed characters. To monitor and accumulate features, a variety of criteria are applied. Features, on the other hand, should be rotation- and size-independent, and computed and selected individually. Offline features were historically characterized as high-level (extracted from words or images), medium-level (recognition and extraction of individual letters), and low-level (recognition and extraction of individual letters) (distinguishing sub-letters) (distinguishing sub-letters) (distinguishing sub-letters). Additionally, feature types may be classed as statistical features, structural features, or features related to global transformations (*Wakahara, Kimura & Tomono, 2001*). The geometric and topological components of a pattern determine its fundamental and special features. Statistical features of an image or image area are numerical metrics (*Xie et al., 2015*; *Sari, Akgül & Sankur, 2013a*). Histograms serve as excellent representations of chain-code directions, pixel densities, moments, and Fourier descriptors (*Sari, Akgül & Sankur, 2013a*). In most cases, statistical features are used colloquially while computing text independently. Meanwhile, structural characteristics are discovered using pattern categorization. For example, the characteristics used in Arabic character identification include various dots, loops, endpoints, overlapping letters, and crossing points and strokes in a range of orientations. Furthermore, the qualities of dots, as well as their positioning relative to the baseline, might be considered structural aspects (*Denton et al., 2014*). In particular, structural aspects are more hard to extract, especially from Arabic text pictures, and different errors may emerge due to minute differences between Arabic letters. The distance among the start and finish pixels in the contouring segmentation phase may impact the $x$-axis, $y$-axis, and structural feature characteristics in contextual text extraction. The training and evaluation materials and associated graphics also describe structural elements such as left, right, top, and bottom directions (*Saber et al., 2017*). Several works (*Obaidullah et al., 2017*; *Roy et al., 2016*; *Dhall et al., 2011*; *Rodríguez & Perronnin, 2008*; *Chherawala, Roy & Cheriet, 2013*; *Terasawa & Tanaka, 2009*; *Lu, Liong & Zhou, 2017*; *Ayyalasomayajula, Nettelblad & Brun, 2016*; *Sun et al., 2016*; *Yuan & Liberman, 2008*) investigate the extraction of characteristics including dot–number connections in individual segments, trains, branching, and secondary strokes (height-to-width and inclines from the initial to the final phase) (*Roy et al., 2016*). Structural factors, such as dot placement, positioning, and diacritic extraction for each ligature, are

also important (*Dhall et al., 2011*) in the offered features. Character length, NUN-position, and the spacing between subsequent lines were identified as structural characteristics of OCR systems in *Ali et al. (2004)*. The statistical analysis for feature extraction is represented in the training phase by an ANN with the smallest average squares (*Mahmoud, 2008*), in which a unique kind of Arabic system was developed utilizing binary digits as input and feature extraction as output. The identification of Persian letters using geometric fitting and Euclidean distances was used to determine the self-esteem for machine operations, with viable transportation matching cover exceeding 98 percent (*Zerdoumi, Hashem & Jhanjhi, 2022*). The most significant part of the Arabic recognition system is cluster. To align patterns with the right characters, data from a number of sources, including handwriting styles, is employed. The following image recognition analysis sample is provided in light of the aforementioned names and phrases. It evolves when the technologies required for each footprint in the account '' succession are used, as well as additional processes such as computer vision, which employs a wide range of complex cursive handwriting to search for hidden values (Fig. 1).

## PROPOSED APPROACHES FOR ZONE-BASED WORD RECOGNITION

This section proposed using an innovative zone segmentation technique for Arabic word recognition, which was built by integrating a hybrid approach based on clustering for Arabic handwriting recognition to arrive at a definitive result. Following the completion of processioning methods, the word features are detected by a section segmentation component, which then identifies the word features. By dividing the word picture into three fundamental zones, the number of critical components that impact character recognition techniques is reduced, which improves the quality of the processing module's output. This is in contrast to traditional Arabic handwritten word recognition methods. deep learning approaches to identify the middle portion components as being part of the middle section after it has been divided into various zones. The SVM classifier predicts the bottom and top zone components, as well as their relative sizes. To get the ultimate result, the results from each zone have to be combined. It is shown that the proposed zone-based system outperforms. The inclusion of zone segmentation to a normal word recognition technique enhances recognition rate (Fig. 2).

In Fig. 2, an Arabic sentence is used to show the various ways. Following that, the complexities of each key phase are briefly covered in the subsections that follow.

### Zone segmentation

Using baseline identification, the letters were separated into three main zones after slanted errors and skew detection were remedied: the bottom zones and the main center zone. For this endeavor, the motivation of the baseline is crucial. You cannot ignore it. It is now safe to go on to the next phase of writing analysis. It's not always easy to come up with the right words. As a result, the identification considers all of the possible segmentation choices. The following description shows the inter-disconnectedness of various strands (Fig. 3).

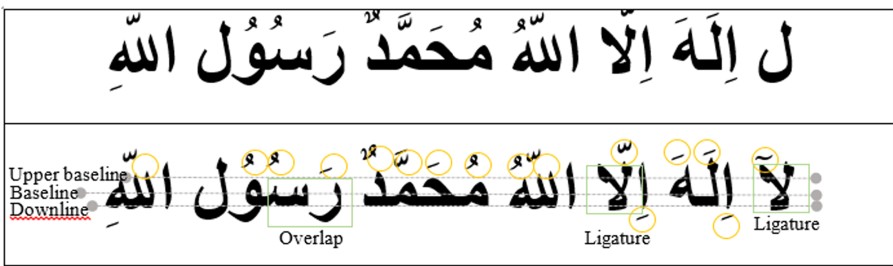

**Figure 2** Top row: original Arabic script (unlabeled). Bottom row: Arabic script with labels marking the upper baseline, baseline, and down line of the Arabic cursive writing.

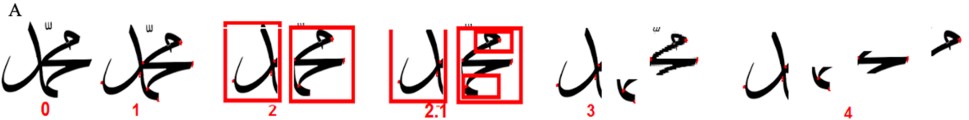

**Figure 3** Steps of zone segmentation-based windows segmentation approach. 0, original image; 1, detection of junction and end points; 2, sliding window; 2.1, detection of ligature (applying algorithms 1 and 2); 3, application of algorithm (baseline detection for splitting the image into upper, middle, and lower zones); 4, segments resulting from baseline detection.

## Baseline detection

Horizontal projections, including the crucial characteristic that regulates the displacement of the baseline, are commonly created using printed word knowledge. The pixels with the greatest peak values in a row are referred to as the line foundation in this context. Purely unstructured handwriting results in an imperfect baseline that includes distorted and fragmented letters. The evaluation of the baseline detection location is based on considering multiple rows of data for limiting and estimating word attributes. We're analyzing the advantages and disadvantages of these three options to see which one is the greatest fit for our situation. The horizontal track's filter projection investigation found that R1, the main row, seemed to have the highest resolution. Our current row R2 was found using the squared Euclidean distance across rows R1 and R2, in which the profundity values were correspondingly lowest.Finally, there is R3, which may be computed as follows:

$$Min(R1.2 - R2.2)^2. \tag{1}$$

Due to the top character region has less articulations, the descending script zone will be as condensed also as bottom and top script region. As a consequence, despite moving from underneath to beyond the baseline, a considerable drop in the estimated peak in the upper half of the features is expected. Whenever the presentation of the main row (R3) exhibited a considerable degradation, the R3 labeling was detected. Finally, the baseline row is obtained using the following norms: 1. depicts Th14H/10 (threshold) and H (deep of phrase features). The depth gradient created by the greatest and low high point pixels in each

paragraph of the cursive characteristics is utilized to produce the H. The preponderance of word aspects rely on this criterion to determine the location of the baseline rows.

## Upper zone segmentation

Subsequently, by predicting the baseline detection, we trace a window directed at the baseline area word extract from baseline (WB) with a height of 4 SS, keeping it in the middle, where it is created Over 90% of the characteristics in the experiment data-set have a curved baseline in the middle word.

After removing the skeleton's major curvatures, couplings, and final points, the word's skeleton is removed. The letter ''P'' was assigned to these sites. We are currently extracting simple shapes connecting future Ps inside the world's middle half, WM. If a stroke fragment arrives from point P with signals WM, it was rejected as a key alphabet component. Single-stroke sections between Ps that pass within WM are considered. If more than one pixel is launched in a certain column, the highest pixel is chosen. In other words, the baseline may have been screwed with or destroyed. In this situation, we used the standard procedure to merge the two nearest pixels. By glancing at the higher regions of the baseline, the adjusters in the upper portion of the baseline may be noticed refer to Fig. 3.

## Lower zone segmentation

It highlights the earlier method. To separate the modifiers in the lower zone, we observe the baseline, which divides the ''middle'' portions from the ''lower'' zone by detecting a character's severe deterioration in zones in the bottom half of the characteristics. This strategy may fail at times, particularly if the baseline capable of separating the features into the lower and middle zones was previously difficult to identify. If the term's letters were unequal in size, and multiple modifiers were present in the lower zones, we may not have seen any boundaries of the alphabets where there was a drop in the prognostication of the lowest pixels among the image's lower and mid-zones. Because of the advanced writing style, the ''bottom part'' noticed in Fig. 3 proves to be challenging to apply utilizing projection analysis. To overcome this problem, our work combines contour matching-based techniques geared at extracting modifiers. We utilize shape matching to detect and segment the modifiers in the bottom half of the image. We detected the tetchy location of the change utilizing skeleton separation and morphological examination of the core zone. If the other contour components coordinated with high comparable confidence through one of the ''lower part'' features, the separated portions would be detached from the central zone. L is defined by the sequence.

Permit middle zone (M) to use the term ''future'' as precisely as feasible. The first element received is M1's skeletal future, and the junction is utilized to determine the end positions of the skeletal features. Assign L1 to be the lower half of M1. The ''Related Component'' (CC) approach is used to distinguish processes that are not connected to M1 as modifiers in L1. Connectivity, according to the CC research, is mostly a gift. The modifiers in the bottom zone make contact with M1. To decouple the modifiers from the lower end points, the component skeleton is submerged. The sensitivity overdue tracing indicates that, in general, entirely lower modifiers must be in the ''lower half'' of the term

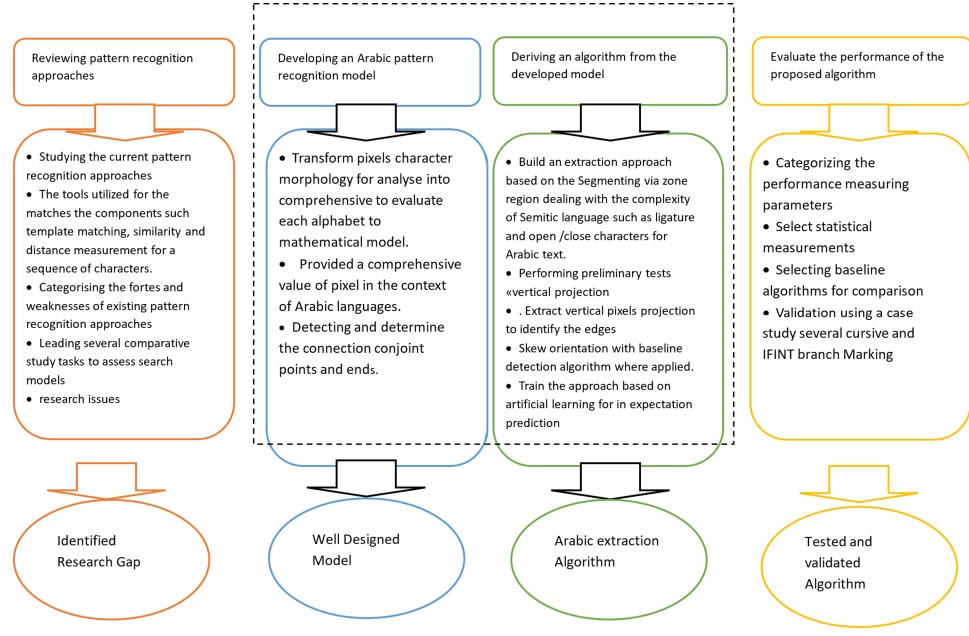

**Figure 4**  **Overall vision of proposed approach.**

features from start to finish, with the exception of ''The Modifier'', which does not need a joint point, as seen in Fig. 3 and algorithm 2.

In most cases, junctions were divided into loops. In terms of condition, we've identified the loop and are sketching the next connecting point for branching off. If a column contained more than one depth point, the deepest endpoint was determined by the segmentation inquiry. After segmenting the features, we use the SVM cluster to compute the recognition confidence and get the class label for the matched feature.

# FEATURE EXTRACTION

In this proposed technique, feature extraction is a significant component of the recognition system, and the center zone is regarded the essential area of Arabic writing, where letters regularly interact. The deep learning used in conjunction with a ''Stochastic Sequential Classifier'' can detect if two components are touching inside this same zone. The difficult construction of a successful feature extraction approach, the pyramid histogram of directed gradients, culminating in the provision of high-resolution HOG features to the a based middle zone recognition system. To prove the frame's utility, four modified state-of-the-art approaches must be implemented and their performances compared to prior work , a local gradient histogram (LGH) was applied to detect the feature throughout the extraction procedure. The research incorporates features, the majority of which are profile features, a GABOR picture, and a G–PHOG image (a blend of Gabor and PHOG) as part of the ''Middle Zone'' or ''Baseline Recognition'' section. These features are covered in further detail in the following sections (Fig. 4).

# EXTRACT VERTICAL PIXELS PROJECTION TO IDENTIFY THE EDGES

To identify the edges, the extracted vertical pixel projection was used. A contour is a shape that describes the fundamental combination that extends the subject characteristics into a localized form through structural configuration, such as the gradient orientation for a certain hierarchy resolution structure. To acquire features with the sliding window, divide the features into many cells at higher levels of the hierarchy. To extract the features *via* the sliding window, dividing the features into individual cells at various hierarchical levels. The grid is made up of 4N4N unique cells, each with N possible solution possibilities (*i.e.*, N140, 1, and 2). (For instance, N140, 1, 2, … ). "The Histogram of Gradient-Orientation" translates the competition for each pixel into single cells quantized into L bins. In the angular radial zone, a single bin correlates to a certain quadrant. There are several of these feature vectors, and they are merged to form "The Descriptor".

The "L-vector" at the zero level represents the "L-bins" of "the histogram" at that level. L*4N*4N "Dimensional-Feature-Vector" L*4N*4N "The Hierarchical Resolution Level" (For instance, N140, 1, 2.) Thus, the final description is as follows:

$$L = p * n * \frac{n}{4} * \frac{1}{4K} \, where \, n \in [1..4].$$

"Boundary-Hierarchical-Level", where D is "Dimensional-Feature-Vector". Apart from examining 8 bins (360° × 45°) of angular data, the key constraint throughout this technique appears to be the level (N) of 2. For the different Sliding Window positions, we obtain (1…8)(4 8)(16 8)14(832128)14168 "Dimensional-Feature-Vector".

## The edges to features

In object recognition, the edges-to-features characteristic, as given by *Chherawala, Roy & Cheriet (2013)*, is still equivalent to the HOG feature (*Terasawa & Tanaka, 2009*). When two continuous frames are overlapped, the right side of a word picture shifts to the left side of the display. In addition, the features were determined by dividing each feature into 44 bacteria for each sliding window. A "Histogram Gradient" (8 bits) was calculated for each cell. To create "128-dimensional Feature Vector" one per consecutive sliding window, the final feature vector binds 16 histograms together. Prior to feature mining, the Gaussian filter was employed to clean up the features to provide the greatest possible gradient learning.

## Features edges to pattern

In alphabets, including word recognition, edges to pattern feature has been employed effectively (*Lu, Liong & Zhou, 2017*). Filters may be found in four different locations, each with a different angle of inclination. It is necessary to utilize the values of 0°, 45°, 90°, and 135° in this context. The magnitude is then used as an acknowledgment for feature extraction in this approach. In order to ensure that such concatenated of almost all of the features in all layer had 48-dimensional properties, the picture features are separated into 12 rows and then piled.

## Pattern to correspondent matching

An outcome depends based on exactly nine features for measuring from the frontal pixels in a single feature column gives a considerably delineated feature for Arabic script. The background pixel's center of gravity section is controlled and regulated by the following order moment, which integrates the three primary global characteristics. District characteristics other than "Dynamic Information" and "Front Pixels" include "Lower and Upper Contour Points", "Foreground to Background Pixel Shift Characters", and "Gradient of Lower and Upper" in the drawing shown in Fig. 4.

## RECOGNITION

The Sliding Window approach is used in combination with deep learning and comprehensive recognition to extract the identification of the intermediate zone by auto-encoder. Feature-Vector-Sequences (*Sun et al., 2016*) are processed from right to left using HMMs with unbroken density. The ability to handle sequential dependencies across prototypes was a game-changer. Deep learning-based features Primary states may be defined by probabilities and the state transition matrix A14 [aij], I j,…,N, where the probability (OK) is represented by a continuous outcome likelihood density and the possibility of evolution from state I to state j is indicated by the value of aij. A specific k-dimensional feature vector, indicated by the function (bj (x)), encapsulates the concentration job. Any state in the model may be treated as a "Detached Gaussian-Mixture-Model (GMM)" in this context. Each state's probability density result should be expressed as follows:

$$bj(x) = \sum_{k=1}^{m1} CjkN\left(x, \mu jk, \sum jk\right). \tag{2}$$

while mj represents the number of Gaussian distributions allocated to j, N (x; ;) represents a Gaussian with a median correlation matrix, jk, and k represents the weight factor of the "Gaussian-Component". The weight of the Gaussian component coefficient of k states to j defines the observations and the state. Take this into consideration as an instance:

If O is an observation sequence O14 (O1; O2, OT) that is assumed to have been molded by a formal sequence Q14 (Q1, Q2, QT) of dimension T, the following estimates the observation possibility or likelihood:

$$P(O, Qj\lambda) = \sum_{Q} \pi q1 bq1(O1) \prod_{T} AT - 1qbq(Ot), \tag{3}$$

where $\pi$ q1 If q1 is true, then There is a sliver of a chance that statistics will occur. The baseline of the word features were untangled using Feature Vector arrangements during the training stage, and other criteria were incorporated. In order to get recognition, the Decoder process was applied. The changes made to the HTK toolkit as a result of *Yuan & Liberman (2008)* are used throughout execution. Constraints that contain "state" and "Numbers of Gaussian Mixture" are safe methods to provide validation data. In the next phases, the images are reduced in size. They are made smaller such that they are 150 pixels wide apiece. *Sun et al. (2014)* utilized the bottom and upper zone modifiers to generate a feature vector that displays how large or little items are. In order for the experiment

to be successful, several qualities were examined. The SVM classifier was employed to categorize these processes (*Tong & Koller, 2001*). Since SVM-Clustering has been shown to be effective in a wide variety of classification scenarios, it was selected for this test. Zone-wise recognition may be trained on data in a data-set termed M. This portion will look at how the combination is actual and how the center zone outcomes impacted the alignment. By using *Kraljevski, Tan & Bissiri*'s (*2015*) "Forced Alignment", a word feature's baseline letters are brought closer to each other, which is why this technique is used in the center zone. Force Alignment were deliberately crafted to display the tiniest alphanumeric boundaries inside the baseline. After determining the bounds of the alphabet in the middle zone, specific borders were applied to the lower and upper zones. This aided with the creation of lower and upper zone alphabets, as well as the association of the primary middle-zone letters. This is one hypothesis to consider about breaking up characters and creating up a new word. Using the N-best listed, we generate N more hypotheses, all starting from the center of the word. The purpose is to generate up with a hypothesis, which involves finding out how much the middle zone was acknowledged. There are two components to this word feature's score: center elements (W) and their later sections (W|X). We can get more out of Bayes' advice by using the logarithm.

$$\log_p(WX)\log_p W - \log_p x. \tag{4}$$

Based on the outcomes of these tests, the N-best hypothesis is chosen. By merging the information from the lower and upper zones (*Hesham, Abdou & Badr, 2016*), the best hypothesis is picked from among these N ideal options:

After calculating the zone-wise appreciation result, the lower and upper zone modifiers of a word are anticipated (the bottom and upper zone modifiers of a word are predicted using deep learning auto-encoder based, and the intermediate zone alphabets are identified using modifiers and overlapped with the word feature [X]). Labels in the middle region which have been largely recognized, connect the lower and upper zones of the alphabet labels. The alphabet labels' connotation revealed a Path-Search difficulty, since each alphabet label was utilized just once to locate the best potential matching word. This is how this combination was presented in this approach:

If N is an arithmetic, then CM 1 through CM N represent the alphabet recognition labels RL's center\zone, and CM N is an integer. Algorithm 3 employs extraction labels for characters in the lower and upper zones.

imminent The accuracy of feature edges to pattern characteristics is 91.21 percent (Arabic) and 92.45 percent (Othmani). To detect the edges, extract vertical pixel projections together. With a "Sliding Window" value of 406 and a phase size of 3, features edges to pattern results are achieved.

Despite the fact that back propagation models may be used to provide installation, mounting is not accessible. Taking into consideration the requirement to be as clear as possible when defining what you mean by "learning", the previous connection of the nodes that interacted with the layers is considered. The regulation of interconnections in deep neural networks, especially in convolutional neural networks, is becoming more recognized as a significant difficulty in the training of architectural layers. There are a

variety of methods for determining how tough a topic is to grasp. To address this issue, one option is to explicitly address it as a problem of supplying appropriate information to lower levels in order for the top layers to create output when it is needed by the lower layers.

Students are able to focus on their strengths and be motivated by their knowledge as they go through the levels, which puts them in the best possible position to succeed. The benefits become clear in each case that is treated individually; all that is necessary is a little ingenuity, for example, while choosing the final layer settings for each situation. We expect to need to use all of the gradient parameters in the regions where we indicated the first layer since the two highest levels of management are now responsible for dealing with the occurrences that were incorrectly classified. Despite this, lower layers would be able to pick up stray particles since over-fitting is not present. If a classically trained model includes parameters that feed forward and are put under-based representations, these parameters will be utilized as classes, but they will ignore the bulk of variance in the nodes. Here's an example of how AN-network encoders may be taught to discriminate between different representations and other things, using the code below:

*Decoder*

$$\hat{x} \qquad = o(\hat{a}(x))$$
$$= \underbrace{sigm}_{\text{for binary inputs}} \left( c + W^* h(x) \right) Encoder$$

$$h(x) = g(a(x))$$

$$= sigm(b + Wx)$$

*Decoder*

$$\hat{x} \qquad = o(\hat{a}(x))$$
$$= \underbrace{sigm}_{\text{APPLIED IN binary inputs}} \left( c + W^* h(\boldsymbol{x}) \right).$$

## EXPERIMENTAL RESULTS

As illustrated in Table 2, the "Handwritten Word Recognition" system was evaluated by gathering two data sets in Arabic, one of which includes the Othmani script. The table displays the data descriptions as well as the amount of words utilized in the tests.

Table 2 contains a description of the data set, including the volume of each word feature for experimental and assessment purposes.

Performance in Arabic cursive writing, including interaction with software-assisted identification of repetitive operations (*e.g.*, Nashua, Riqaa, and comparable scripts written

**Table 2  traing dataset\validation\test set.**

|  | Training dataset | Validation dataset | Testing dataset | Total |
|---|---|---|---|---|
| Arabic script using IFNT | 12,253 | 2,982 | 1,856 | 17,091 |
| Othmani script | 12,667 | 3,872 | 3,589 | 16,128 |

for the Holy Quran) reefer in to Fig. 5. Arabic letters are difficult to decipher because of their overlapping and interlocking nature. Syntheses Learning-based sequence modeling is utilized to solve the difficulties of character segmentation in Arabic cursive writing. It also provides an efficient Arabic calligraphic word identification approach that divides words horizontally into bottom, middle, and upper parts, and then identifies them based on their different parts, including the difficulty of tachikile. The zoning is designed to lower the total number of classes in Arabic cursive writing by reducing the number of different element classes. It has been discovered that following the suggested technique increases the likelihood that people would recognize the approach. Alphabets tend to be tender and easily identifiable at the mid-zone (baseline) of the segmentation zone. As a result of the conjoined zones' identification, right and left characters' alignment is given extra credit. A cursive border emphasizes any remaining features in the bottom and upper zones, if any are there. The character's morphology and the alphabetic elements are then joined to form a full sentence. Recognition at the word level Zone segmentation, especially in the upper zone of a character, relies on these fundamental attributes to better spot border faults during the segmentation step. This is what we want to do in the future. Zone Segmentation Methods for Arabic Word Recognition It is suggested to observe with software interaction in order to accomplish lower and higher zone recognition. In a comparison with other features of a similar kind, this feature was shown to be reliable for Arabic handwritten script recognition. Tests on different hand writings utilizing additional assets such as IFN and IFNT were undertaken to evaluate the recognition performance. Zone-wise recognition beats the existing technique to word recognition in terms of accuracy, according to these research. Allows for a sensation of an extract feature superimposed on calligraphy without the need for distinct explicit proportions between the letters. ''ZONE-WISE'' experience structure and approach Using the frame and scheme, you may draw characters with a single stroke or a series of them. Cursive calligraphy may also be examined on its own. An input signal or the dialect's qualities and attributes may be used to determine where information words are located in the context of a given language. Discontinuity, division, knowledge of roasting function, and a visual representation of dialect are all used in this technique to deconstruct the role of calligraphy. At the very least, many procedures will be carried out at the same time. Over 17,091 handwritten word features in Arabic scripts were gathered. These characteristics were culled from sixty diversely worded texts authored by people from a variety of occupations. A total of 11,253 of these word features are for validation, while the remaining 3,856 are for testing purposes. Arabic words were officially acknowledged as part of the English language in 1982. We collected all 16,128 handwritten features for the Othmani script, of which 10,667 word pictures were used in the application step, 1,872 images were used in the validating stage, and the remaining 3,589 features were used in

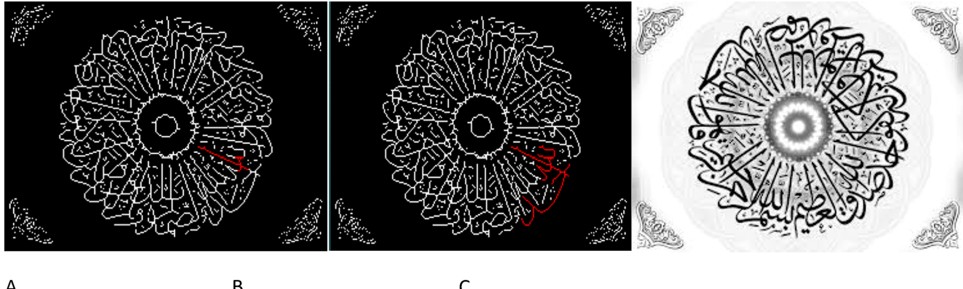

A            B            C

**Figure 5** **Complex zone segmentation for extracting from Othmani scrpt: (A) original image; (B) detection of junction, end points and sliding window; (C) detection of ligature (application of algorithms 1 and 2).** The figure demostrated the process of employing an advanced deep learning technique to extract complex segments from Othmani script. This involves a three-phase approach denoted as A, B, and C, representing the original image, junction and endpoint detection along with a sliding window, and the identification of ligatures using Algorithms 1 and 2, respectively. The framework aims to perform zone segmentation, allowing direct perception of Arabic word characteristics from calligraphy without necessitating explicit character proportion representation. It enables processing of characters in single or multiple strokes and handles cursive calligraphy independently. Moreover, this strategy defines word boundaries by either utilizing specific shape input signals or recognizing limits based on dialect attributes, dissecting calligraphy through discontinuity, division, function recognition, and dialect visualization, possibly occurring simultaneously. The input comprises Arabic script/Quran in the Othmani style, extracting compressed features and converting them into text values for authenticating Quranic verses using a string matching algorithm proposed by the research team. The recognition system interface (Output (string value)) distinguishes Arabic and Quranic text values, crucial for verse authentication, aligning with research outlined in *Hakak et al. (2017)*.

the testing stage. Table 3 illustrates the detachments from this dataset. For extra study, the Arabic and Othmani script datasets are accessible online at *Dharani & Aroquiaraj (2013)*. 3. Figure 3 illustrates the distribution of words by character expansion. Four-character words were rewritten according to the Arabic dataset's widest range and writing position.

After per-processing for each word's features, the middle zone is recognized. The sentence is divided into three different zones by horizontal zone segmentation. The medial zone is similar to that which was treated with an HMM-based technique. The vocabulary's size in

Because of the approach's usage of zone segmentation, the recognition of the mid-zone is reduced. By using zone segmentation, we were able to produce 1,518 and 1,894 hybrid words that were formed from plain Arabic and Othmani scripts, respectively, by using zone segmentation. We estimate the number of HMMs that occur successively in each state using diagonal co-variance models of the GMMs for each state (*Hakak et al., 2017*). The five primary differentiating characteristics are provided by the preliminary representations. Using a Gaussian-Mixtures symbol, the validation data from both data-sets checks the state numbers in each database. As a result of zone splitting, in their respective data-sets, the sign of the Arabic and Othmani scripts' alphabet class into HMM has been reduced from 124 to 42 and 116 to 40 pixels, respectively, as a result of the deterioration. shows the performance of the top N options as well as a variety of Gaussian mixes based on data from the validation study. In our approach for determining N ideal choices, we employ

**Table 3** The cross validation for similarity/recognition for each Arabic individual including the cross characters.

| | ا | ب | ت | ث | ح | خ | ج | س | ش | ص | ض | ر | ز | ط | ظ | ك | م | ن | ف | ق |
|---|---|---|---|---|---|---|---|---|---|---|---|---|---|---|---|---|---|---|---|---|
| ا | 1 | 0 | * | 0 | * | * | * | * | * | * | * | * | 0.05 | * | * | * | * | * | * | * |
| ب | * | 1 | * | 0.09 | * | * | * | * | * | * | * | * | * | * | * | * | * | * | * | * |
| ت | * | 0.16 | 1 | 0.15 | * | * | * | * | * | * | * | * | * | 0.02 | 0.01 | | 0.01 | 0.34 | 0.2 | * |
| ث | * | 0.13 | 0.7 | 0.82 | * | 0.01 | | * | * | * | 0.06 | * | * | 0.04 | 0.03 | 0.6 | * | 0.28 | 0.3 | |
| ح | 0.3 | * | * | 0 | 1 | 0.32 | 0.1 | * | * | * | * | * | * | * | * | * | 0.01 | * | * | * |
| خ | 0.4 | * | * | 0 | 0.32 | 1 | 0.9 | * | * | * | * | * | * | * | * | * | * | * | * | * |
| ج | 0.25 | * | * | 0.01 | 0.01 | 0.02 | 1 | * | * | * | * | * | * | * | * | * | 0.01 | * | * | * |
| س | * | * | 0.26 | * | * | * | * | 1 | | 0.19 | * | * | * | * | * | * | * | * | * | * |
| ش | * | * | * | * | * | * | * | 0.03 | 1 | 0.1 | 0.70 | * | * | * | * | 0.02 | * | * | * | * |
| ص | * | * | * | * | * | * | * | 0.01 | 0.01 | 1 | * | * | * | * | * | * | * | * | * | * |
| ض | * | * | * | * | * | * | * | 0.02 | 0.01 | 0 | 1 | * | * | * | * | * | * | * | * | * |
| ر | * | * | * | * | * | * | * | * | * | * | * | 1 | 0.05 | * | * | * | * | * | * | * |
| ز | 0.11 | * | * | * | * | * | * | * | * | * | * | 0.04 | 0.93 | * | * | * | * | * | * | * |
| ط | * | * | * | * | * | * | * | * | * | * | * | * | * | 1 | * | * | * | * | * | * |
| ظ | * | * | * | * | * | * | * | * | * | * | * | * | * | 0.7 | 0.97 | * | * | * | * | * |
| ك | * | * | * | * | * | * | * | * | * | * | * | * | 0.02 | * | * | * | 0.99 | * | * | * |
| م | * | * | * | * | * | * | * | * | * | * | * | * | 0.01 | * | * | 0.01 | 1 | * | * | * |
| ن | * | * | * | * | * | * | * | * | * | 0.01 | * | * | * | * | * | * | 0.2 | * | * | * |
| ف | * | * | * | * | * | * | * | 0.01 | 0.01 | * | * | * | * | * | * | * | * | * | * | * |
| ق | * | * | * | * | * | * | 0.01 | * | * | * | * | 0.01 | * | * | * | * | * | * | * | * |

an algorithm to fuse the upper and lower baselines with the applicable Arabic pattern, which results in N optimal picks. A total of 32 gauges are used in this combination. It has a jagged design on the outside.According to the validation data, the features generate outcomes with up to 92.98 percent accuracy for Arabic script and the top five elections. Descriptors exceed other characteristics in the baseline in terms of accuracy. To detect the edges, a vertical pixel projection is used. From pattern edges to pattern outcomes, a "Sliding Window" measurement of is used to get the desired results. Besides combining the property of rigorous matching with the cross validation function, the cross validation function gives an overall image of the system. The ratio of similarity and recognition for each Arabic individual character will be taken into consideration in Table 3 if there is a text character with a maximum number of abstract qualities associated with it. The intention of the Table 3 cross-validation is to test the capability to expect new Arabic features that were not utilized, including those not involved in the per-estimating phase. This strength needs to be deal with issues of problems over fitting in order to avoid the bias of model performance in ideal cases only in selected features selection bias; for instance, the alphabet simile stockers, including the morphological with other alphabets with only 0.25 overall, which is considered lower than the average with 0.51 recognition, to take it into

consideration as similarity, besides taking it to the next level of recognition to deliberate as matched alphabets.

On the other hand, the Arabic alphabet that has the dots above the baseline, by providing the ration perimeter not exceeding 0.3 as an angle value (the features in the experiment protocol in this case), can provide a very confident recognition rate. Despite the fact that the serial dots in handwriting are above the baseline in alphabets, the recognition in this case could be considered as higher similarity adjacent to the exact match to this instance of handwriting with only 0.6 recognition rate, and the system container appropriated it as one of the best explanations if the backmost move forward to find the maximum value to consider as an optimum solution.

In this case of the exact match, the considered is has a rate of 100%, which is the extract with itself. With this rate of matching, in addition to the identical rare cases, the same features validation table set of the same data is used as branch marking or as the main reference to identical the text value with its associated set of features.

In Table 3, the ratio for similarity comprehends clearing the diagonal matching with an approximated precise match ranging from 0.81 to 1 of the probable identical

The estimation of the range of test features is pursued to validate the numerous similar images to fit and model validation for measuring the manner of the result in statistical analysis, simplifying it to an autonomous data set for further prediction.

The validations in Table 3 are focused on the main recycled in scenes where the project's goal is to precisely approximate and perfect the projection for further prediction in practice

The potential issue with this proposal. The Arabic model appears in the prediction of old cursive, where the model has various features besides the very limited data-set such as branch marking used in the field. To solve this issue, they selected old cursive writing to be re-added into the Arabic context, given a data-set of known data on which to prepare features as part of setting experiments (training data-set). To stretch a vision, the prototypical is distilled to an autonomous data-set (*i.e.,* an unidentified data-set).

One example of cross-validation comprises dividing an illustration of features obsessed *via* supported additional subsets and evaluating the examination on one of the training sets.The validation set or testing set is used in the examination, and it contains 67 percent and 33 percent of the amounts of the features given in the training stage, respectively. Different feature divisions are then unfilled to decrease inconsistency and input the validation result. For the Arabic training data-set, the identification rate for upper changed words was totally made up of 1,156 lower zone and 1,223 upper zone alternates, whereas the Othmani script included 937 upper and 851 lower zone . This approach provides 500 alternates for testing in various areas to evaluate the implementation. Table 4 displays slices of the traits discovered during study.

Table 4 displays the qualitative findings of SVM application. By integrating the zone-wise layouts, complete word recognition is obtained. Then, from the three zones, zone-wise sequences are produced and connected to make the entire word. The incorporation involved utilizing a mapping function specifically designed to connect the segments within the mid-zone.

**Table 4  The Mid-zone recognition outcome using different word features dimensions.**

| The Script | Accuracies with respect of word dimension | | | | | |
|---|---|---|---|---|---|---|
| | 2 | 3 | 4 | 5 | 6 | 7 |
| Arabic script | 72.19% | 66.31% | 79.36% | 61.25% | 66.21% | 63.14% |
| Othmani script | 88.65% | 89.19% | 74.21% | 76.78% | 79.79% | 97.45% |

**Table 5  The SVM cluster results of lower and upper modifiers.**

| The Script(s) | The Modifiers | The Training dataset | The Testing dataset | Accuracy (%) | |
|---|---|---|---|---|---|
| | | | | Upper 1 | Upper 2 |
| Arabic script | Upper zone | 1823 | 500 | 87.66 | 97.23 |
| | Lower zone | 1937 | 500 | 84.07 | 95.15 |
| Othmani script | Upper zone | 156 | 500 | 85.95 | 96.52 |
| | Lower zone | 151 | 500 | 91.94 | 98.14 |

Table 5 summarizes the findings Ci14F (CM i, CU i, CU i 1, CU i 1, CL i, CL i 1, CL i1) is used to derive the upper and lower zone counterparts.

Additionally, the distance between us is used to get the best feasible result for the provided component lists and to study the word's lexicon. Several features of advanced full-word recognition occurrences are shown in Fig. 4. Certain words need particular formatting due to their variable slant angles and skewness. Additionally, word recognition is complicated by the lower and upper alternates. These words are extraordinarily well understood when approached in the manner described. At the full-word level, Table 6 illustrates the features of successful recognition. In Arabic script, we get an accuracy of 83.39 percent from left to right and 92.89 percent from top 1 to top 5 selections. Between the top 1 and top 5 selections, the Othmani cursive has an accuracy rate of 84.24 percent and 94.51 percent. Table 5 indicates how the suggested word recognition algorithms improve with each step. $h(x)$: the hidden layers function in order to urge the output layer to be as perfect as the value in the vector X backward layer's input value, and to keep track of all pertinent facts. If the z = size of the encoder function is smaller than the decoder and will compress the information. Almost every problem may be solved with multi-layer neural networks. While accurate data classification is important, such as the shapes and features of Arabic characters, character qualities may be developed by layering them on top of one other. In principle, training neural networks with several hidden layers should be straightforward, but in fact, it is a complex process. Training one layer at a time is a great way to improve the overall efficiency of a neural network with several layers. To construct an auto encoder network, you must first choose a network architecture for each hidden layer. Using just the information accessible to it, this model demonstrates how a neural network with two hidden layers can be equipped to classify the features into comprehensives shapes *via* only the information available in interconnected between the layer. The first and most important thing you do is practice the hidden layers in an uncontrolled environment using auto encoders.

**Table 6** Display of the accuracies of entire phrase recognition via the feature into the middle zone.

| The Script(s) | Recognition accuracies | | | | |
|---|---|---|---|---|---|
| | Upper 1 | Upper 2 | Upper 3 | Upper 4 | Upper 5 |
| Arabic script | 83.39% | 87.75% | 89.67% | 91.41% | 92.89% |
| Othmani script | 84.24% | 89.14% | 91.47% | 92.18% | 94.51% |

**Table 7** SVM classification results of lower and upper zone modifiers.

| The Script(s) | The Modifiers | The Training dataset | The Testing dataset | Accuracy (%) | |
|---|---|---|---|---|---|
| | | | | Upper 1 | Upper 2 |
| Arabic script | The upper portion | 1923 | 500 | 87.66 | 98,23 |
| | The lower portion | 1937 | 500 | 88.07 | 96.15 |
| Othmani script | Upper portion | 156 | 500 | 88.95 | 96.52 |
| | Lower portion | 151 | 500 | 93.94 | 98.14 |

To complete the training process, the final supervised network is linked and taught to soft max layer, which is then disconnected. In forecasting or reconstructing functions associated with the original input of the victor X, the class label will be suitable, and for this regards, we will refer to the original input as $X^\star$.

## Recognition of the middle zone

Figure 3 illustrates the differences in the two scripts' treatment of the central zone while dealing with five distinct qualities. Several word properties have been identified as full features for the purposes of lexical analysis. A few examples of central zone identification issues handled using the PHOG function and N peak selections are shown in Table 7. Using lower and higher zone transformers, specific choices are then thoroughly cleaned to get an accurate outcome. As the length of an expression increases, so does the identification performance of several PHOG properties, as shown in . Words now include a greater variety of alphabets, a sign that the accuracy of word verification has increased.

## The lower and Upper zone recognition arrangements

For the Othmani script constructed *via* alternative recognition, the training Arabic data-set comprises 1,223 upper zone transformers and 1,156 lower croppers, for a total of 2,390 upper position modifiers and 1,788 lower position modifiers in the upper zone and lower zone, respectively. It is estimated that there are 1,223 upper zone transformers and 1,156 bottom croppers in the training Arabic data-set. The recommended approach selects more than 500 transformers for testing at each of these locations, with the results being analyzed (*Abdelaziz, Abdou & Al-Barhamtoshy, 2016*) after they are tested.

Table 7 presents a summary of the results of the output's investigation of what been seen in Fig. 4, which elaborated that the qualitative outcomes gained.

**Table 8  The comparative training of successive improvements of entire word recognition accuracies including employment of novel recommended technique.** (ΔIUZ is upgrading in Upper segmentation via the principle, ΔILZ is enhancement in lower zone segmentation via shape matching approaches, ΔIC is improvement in combination of zone results using refined character alignment.)

| Method | Arabic | | Othmani | |
|---|---|---|---|---|
| | Upper 1 | Upper 5 | Upper 1 | Upper 5 |
| Zone segmentation approach | 80.21 | 90.87 | 81.31 | 91.94 |
| Zone segmentation approach+ΔIUZ | 81.77 | 91.25 | 82.64 | 93.11 |
| Zone segmentation approach+ΔIUZ+ΔILZ | 82.89 | 92.34 | 83.78 | 94.05 |
| Zone segmentation approach+ΔIUZ+ΔILZ+ΔIC | 83.39 | 92.82 | 84.24 | 96.51 |

# COMPARISON STUDY

To run an external assessment of the experiment, we utilized the most frequently used complete word recognition method combined with indeterminate location segmentation. In the database utilized for performance assessment, a non-zone segmentation-based technique is employed. By patterning the edges of sliding windows, the following sliding window-based features are generated, which are then used for deep learning-based identification. The database's one-of-a-kind effects are the product of well calibrated constraints. According to Table 7, our outer zone segmentation approach achieves an efficiency of 98.23 percent (Arabic) and 98.14 percent (Othmani), while our location segmentation-based technique achieves an accuracy of more than 87.66 percent (Arabic) and 91.94 percent (Othmani). Thus, zone segmentation may continue to be widely used in the future. In comparison to previous methodologies, using an SVM-based holistic strategy and an aggregation of parameters for Othmani cursive identification produces remarkable results. For issues with 50 and 100 features, respectively, the efficiency rates were 81.14 percent and 84.02 percent (*Xu, Wang & Lu, 2016*). I used a technique called "Adapted Quadratic Discriminant Function Classifier-Based Dynamic Programming" (QDFCBD programming) to identify Mediterranean words (cities ranging from Algeria to Tunisia) (*Jayech, Mahjoub & Amara, 2016*). Because it was created expressly for Othminic script, word four has no counterpart in the proposed system. Over 98 percent of Othmani and Arabic script poems have been translated properly into 84 and 117 verses, respectively (see Tables 7 and 8). In this study, we developed an advanced classifier algorithm for Arabic word recognition. To validate the system, we use the well-known IFN/ENIT data-set. The work is of exceptional quality. By combining SVM and HMM classifiers, the number of false positives is reduced. We want to utilize training data to teach the computer its true class for every combination of classifier outputs. Our strategy was validated by the rubrics created by combining the classifiers. We compute the average in the first phase, and then the average weighted by the recognition rates achieved by the classifiers during the training phase. Around 1,500 pieces were utilized to measure the model's rate of achievement and recognition. With a score of 96.51 percent, this statistic recognizes classifiers who have a strong grasp of the classes. Compression may be used in a variety of ways, as seen in Table 9.

**Table 9** The test example of synthase learning approaches for recognizing arabic word features via zone segmentation performance including the viewing with interaction of software associated.

| Word/features | إسم | الله | الرحمن | أبوبكر | البلد | السعودية | عزام |
|---|---|---|---|---|---|---|---|
| الله | − | + | − | − | − | − | − |
| الرحم | − | − | +/− | +/− | −/+ | − | − |
| الرحيم | − | − | − | +/− | − | − | − |
| قل | − | − | − | − | − | − | − |
| بسم | +/− | − | − | − | − | − | − |

**Notes.**
+ matched the word with very high recognition rate above 80%.
− Totally unmatched.
+/− fairly matched due the similarity of skotlone of character but still below 50% of recognition.

The field of "global optimization" investigates methods that locate the Arabic words matches as best comparative solution to a certain issue. Algorithms such as simulated annealing and evolutionary methods global search algorithms. Algorithms developed for certain non-convex problems (which may be discrete) like matrix/tensor factorization. Returning back to core issue of the Arabic language that nature of Geometer witch is rely on recognizing basic elements such as the surface's curvature, which is solidly descriptive, rather than on identifying complicated qualities due to the obvious complexity of interpreting Arabic features. The exhibit features a feature called as "mode connection", which has been discovered. Many nodes have been shown to be a (nearly) equal-value route connecting two global minimal, which were discovered independently of one another. A "global minimum" is a low-error solution that may be achieved by training from two random starting points. A select and per-training technique may be used to "connectivity of sub-level sets" optimization. On the bottom baseline, the charts JIME and HAA both depict the same phenomenon. It is useful to integrate two global minimal through equal-value routes if the sub-level set connectivity function, known as cF, is connected. Using the 1-hidden layer as an example, the link between the sub-level sets was first established, in order to assess the connectivity based on an experimental setup. Use of path-finding algorithms has had a major impact on the field. Path-finding methods to verify the relationship between database global minimal.

*As instance:*

$$tested\,features = \begin{pmatrix} alif_0^{(1)} & alif_0^{(2)} & \dots & alif_0^{(m-1)} & alif_0^{(m)} \\ baa_1^{(1)} & baa_1^{(2)} & \dots & baa_1^{(m-1)} & baa_1^{(m)} \\ \vdots & \vdots & \vdots & \vdots & \vdots \\ yaa_{12286}^{(1)} & yaa_{12286}^{(2)} & \dots & yaa_{12286}^{(m-1)} & yaa_{12286}^{(m)} \\ x_{12287}^{(1)} & x_{12287}^{(2)} & \dots & x_{12287}^{(m-1)} & x_{12287}^{(m)} \end{pmatrix} (feautrestobextracted)$$

$$= \begin{pmatrix} FEX^{(1)} & FEX^{(2)} & \dots & FEX^{(m-1)} & FEX^{(m)} \end{pmatrix}$$

$$matchedmatrix = \begin{pmatrix} matchedalif_0^{(1)} & matchedalif_0^{(2)} & \dots & matchedalif_0^{(m-1)} & matchedalif_0^{(m)} \\ baa_1^{(1)} & baa_1^{(2)} & \dots & baa_1^{(m-1)} & baa_1^{(m)} \\ \vdots & \vdots & \vdots & \vdots & \vdots \\ siin_{12286}^{(1)} & shinnx_{12286}^{(2)} & \dots & baaffa_{12286}^{(m-1)} & xu_{12286}^{(m)} \\ x_{12287}^{(1)} & x_{12287}^{(2)} & \dots & x_{12287}^{(m-1)} & x_{12287}^{(m)} \end{pmatrix}$$

$$matchedmatrix = \begin{pmatrix} y^{(1)} & y^{(2)} & \dots & y^{(m-1)} & y^{(m)} \end{pmatrix}.$$

The assorted between the tested feature and matched feature is defined by the rules of cluster that will be determine in features extraction and matching by shortly noted as Assumption (matched matrix : ^testedfeatures) the demonstration of the model will be explained in the figure followed Conceptual of the optimal connection of Arabic characters, spread out the different traits of character.

## DISCUSSION

The findings are discussed by addressing the following research questions: First, the specifics for constructing an Arabic pattern recognition model, such as improvement, the poor accuracy of Arabic-OCR systems by directing a novel Syntheses Learning Approaches for Recognizing Arabic Word Features toward segmentation.The recognized those major difficulties, a few of those were mentioned as a result of the audit the technique and the evaluation section structure and software.The demonstrations are sketched, and their responses are finally analyzed. Based on the reported processioning square abatement, the recognition accuracy should be reliable. There are several irrelevant features. Similarly, both preparations include testing samples, which reduce the credit precision derived from the created model.

(1) A novel descriptor applied in Arabic text identification in context of resistant to scaling,font type, rotation, with font size changes has been created by solving the original research question?

(2) The domain will progress as long as there are a variety of distinct approaches for building an Arabic text descriptor that meets the criteria for Arabic text identification while also being resistant to changes in scaling, rotation, type of the T3 font, including font size.

(3) The goal is to create an automatic system for identifying handwritten writing. Strategies that have proven effective over the course of many years are zone-wise techniques. Using the results of the skeletons investigation, the whole phrase detection accuracy for dependent emphasis identification in both Othmanic and Arabic cursive is shown in Fig. 6 for both languages. Because of the unusual character structure of Arabic cursive writing, segmentation and categorization are difficult tasks. This poses a variety of difficulties, such as the impossibility to create letters that overlap or are too close together (*e.g.*, Naskha, Riqaa, and other analogous scripts developed for the ancient cursive).

(4) The evaluation focused on two critical elements in a recognition system: extraction operations and classifier combination.

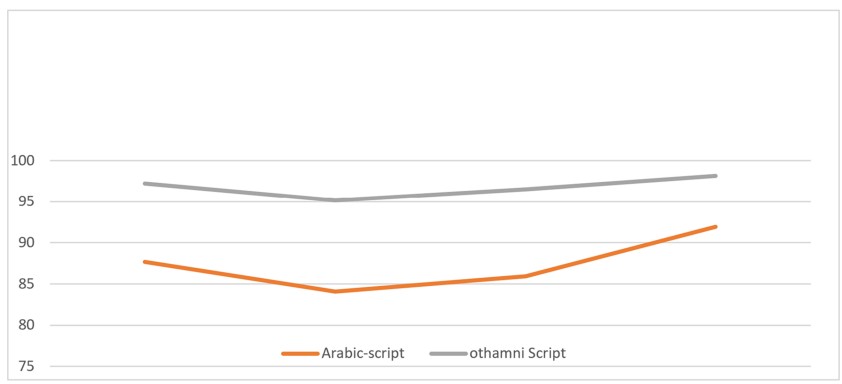

**Figure 6  Plot of SVM classification results of lower and upper zone modifiers.**

(5)  Instants were utilized as feature extractors in such phase, and they were successful at recognizing Arabic characters. On the way to being public and standard database-ready, as seen in Fig. 6, is the intended description's implementation and validation.

This is being accomplished by answering the key research question: how can a new descriptor for Arabic text identification be created that is unaffected by scaling, rotation, font type, or font size changes?

The goal is to create an automatic recognition system for handwritten scripts; experiments will be more fruitful if several novel methods for developing an Arabic text descriptor are available that meet the requirements for Arabic text detection while remaining robust to angular velocity, scaling, font type, and font size variations. Zone-Wise and other techniques may be effectively coupled. Figure 6 illustrates whole phrase recognition for dependent accented recognition rate for both Othmanic and Arabic cursive, as well as the dependent accented recognition rate. Because of the intricacy of Arabic characters, segmentation and identification challenges in Arabic cursive writing are unique (as in Naskha and Riqaa, among other comparable scripts developed for the ancient cursive). Worries regarding writing overlapping and close characters, as well as concerns about writing overlapping and close characters, are all mentioned.The research focuses on two crucial aspects in the development of a recognition system: extraction operations and classifier configuration.These stages function as feature extractors and produced astounding results when it came to detecting Arabic characters. Figure 6 demonstrates the progression towards establishing and validating the proposed description, ensuring its universal applicability across public and standardized databases. According to the comparative analysis shown in Table 10, the system's accuracy would improve as a result of the input characteristics' superiority. A well-chosen data-set reduction strategy contributes to an increase in the total processing rate (it upsurges the system recognition rate). The technique of feature extraction, which is guided by the assignment of Arabic features, often delivers the enhanced result directly from the input photographs. When we examine the input image, the term "Dimensional reduction" immediately comes to mind. In Fig. 7, the

**Table 10  Semi-continuous 1-HMM HMM (one for each word). Comparative analysis of continuous HMM, multi-Classifiers (ANN, SVM, PGM), and hybrid approaches for classification.**

| Author(s) | Classifier | Data | Accuracy |
|---|---|---|---|
| *Pechwitz, Maergner & Abed (2006)* | semi-continuous 1-HMM | IFN/ENIT | 89.1% |
| *Khorsheed (2007)* | HMM (one for each word) | 32,000 words from manuscripts | 85% |
| *Azizi et al. (2010)* | Multi-classifiers (SVM, K-NN, ANN, HMM) | 10,000 words | 93.96% |
| *Haboubi et al. (2009)* | ANN | 16,107 images from IFN/ENIT | 87.1% |
| *Chen et al. (2010)* | SVM | PAWS 7346 | 83.7% |
| *Khémiri, Kacem & Belaïd (2014)* | PGM | 5,279 WORDS | 92.19% |
| Our system | hybride SVM/HMM | From IFN/ENIT | 94.51 |

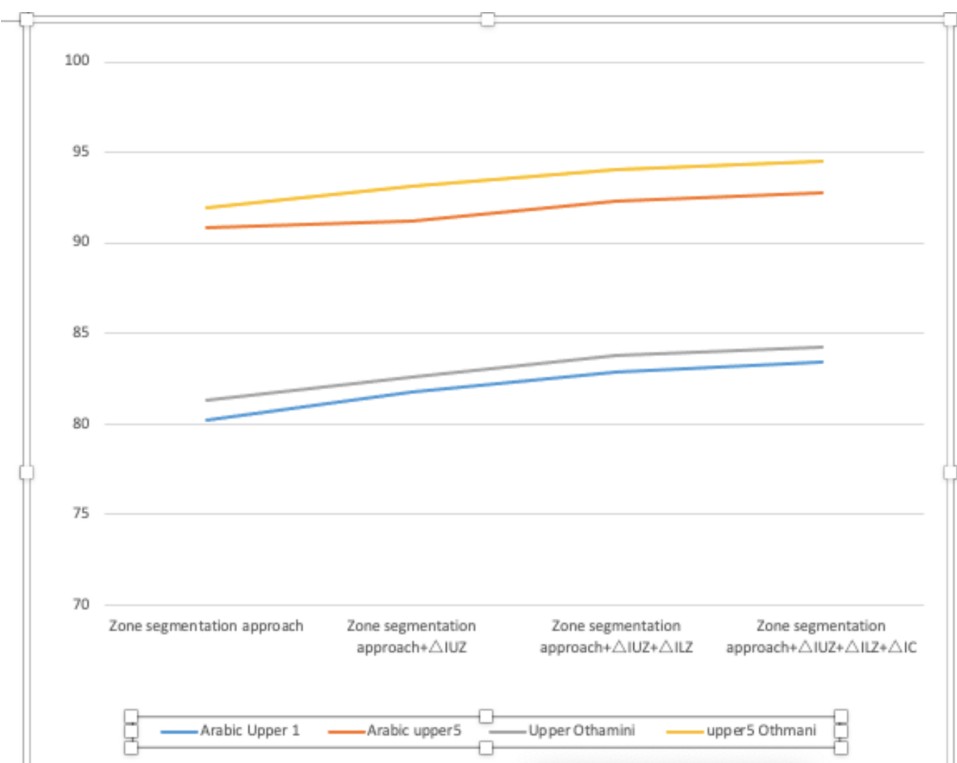

**Figure 7  The comparative training of successive improvements of entire word.**

cumulative accuracy of removing unsuitable features decreases as the overall processing time decreases.

Categorize alphabet {القحابتس} according to their basic components (lower, middle, and upper). Zoning is meant to limit the number of distinct element classes in Arabic cursive writing in comparison to the overall number of classes. This raises the recognition rate of the method. Components of the segmentation zone are recognized, notably in the mid-zone (baseline), which is often where alphabets are tender. The recognition of the center zone imbues the alignment, especially from the right and left characters into the conjoined zones, with a powerful mark. After that, any remaining components in the

bottom or upper zones are highlighted with a cursive border.The alphabets that include components of the Arabic tackle (-) must be included since they are combined with the character's morphology to accomplish whole-word recognition.

Our fundamental property is included into this approach by optimizing zone segmentation performance, particularly for a character's upper zone, in order to identify boundary detection imperfections during the segmentation stage (refer to Fig. 7 and the different training sections for providing good quality of segmentation and its relation with Table 10: Deep learning-based approaches for detecting Arabic word characteristics using zone segmentation).

## CONCLUSION

This article provides a unique method to deep learning techniques for recognizing Arabic word features using zone segmentation performance, which includes interactive viewing of software-assisted identification of repetitive movements in Arabic cursive writing (*e.g.*, Nashua, Riqaa, and related) (*e.g.*, Nashua, Riqaa, and related).Written for the Quran, these scripts Arabic characters, which feature characters that overlap and touch, have a complex complexity. A complete sequence modeling technique based on syntheses learning is employed to handle these character segmentation difficulties in Arabic cursive writing. Additionally, this article proposes an efficient approach for Arabic calligraphic memorizing vocabulary by horizontally segmenting the handwritten word features into three main segments (lower, middle, and upper), and later recognizing them by their correlating segments, including the problematic of Tachikile. It is hoped that the zoning would reduce the total number of Arabic cursive writing courses to a manageable quantity. The recommended technique is found to boost the approach's recognition rates. There are several parts of the segmentation zone where letters are soft and discernible, especially in the mid-zone (baseline). By recognizing the center zone, alignment obtains a huge boost, most notably from the right and left characters into the conjoined zones. Following that, if there are any residual components in the lower or upper zones, they are highlighted with a cursive border. The character's morphology and the basic alphabetic components are then merged to generate the final product. Recognition at the word level. In order to increase zone segmentation performance, especially in the top zone of a character, and discover border detection flaws during the segmentation stage, these key attributes are included into their approach.This is where we are headed. Using zone segmentation to recognize Arabic word features, an upper and lower zone recognition performance, as well as visual display and software interaction, is being offered. This attribute was demonstrated to be strong for Arabic handwritten script recognition in a comparative evaluation with other related features. To assess the recognition performance, comprehensive tests on varied handwriting are undertaken using a range of data-sets, including IFN and IFNT. The results of these investigations reveal that the recommended zone-wise recognition strategy achieves a superior degree of accuracy than the presently employed word recognition approach.

### Funding

The authors received no funding for this work.

### Competing Interests

Noor Zaman Jhanjhi is an Academic Editor for PeerJ. The authors declare there are no competing interests.

### Author Contributions

- Saber Zerdoumi conceived and designed the experiments, performed the experiments, analyzed the data, performed the computation work, prepared figures and/or tables, authored or reviewed drafts of the article, and approved the final draft.
- NZ Jhanjhi performed the experiments, prepared figures and/or tables, authored or reviewed drafts of the article, and approved the final draft.
- Riyaz Ahamed Ariyaluran Habeeb performed the experiments, prepared figures and/or tables, authored or reviewed drafts of the article, and approved the final draft.
- Ibrahim Abaker Targio Hashem performed the experiments, prepared figures and/or tables, authored or reviewed drafts of the article, and approved the final draft.

### Data Availability

The raw data and code are available in the Supplemental Files.

### Supplemental Information

Supplemental information for this article can be found online at http://dx.doi.org/10.7717/peerj-cs.1465#supplemental-information.

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
