# Peer review of "A deep learning based approach for extracting Arabic handwriting: applied calligraphy and old cursive"

_PeerJ Computer Science, doi:10.7717/peerj-cs.1465_

## Round 0.1 · original submission · Minor Revisions

The authors are advised to do minor grammatical errors and improve the literature survey. Please address all of the reviewers' comments.

Reviewer 1 ·

Basic reporting

The authors presented a better work entitled, A deep learning for segmentation based on baseline approach for extracting Arabic Handwriting: Applied Calligraphy and Old Cursive.
The authors claims that According to the findings of this research, a novel technique for segmenting Arabic offline text is presented, in which the core splitter of the “Middle” portions from the “Lower” zone is identified by detecting character sharps degeneration in zones. With the exception of script localization and the essential feature of determining which direction a starting point is pointing, the baseline also functions as a delimiter for horizontal projections. Despite the fact that the bottom half of the characteristics is utilized to differentiate the modifiers in zones, the top half of the characteristics is not. Especially effective when the baseline is capable of separating characteristics into the bottom zone and the middle zone in a complex pattern where it is difficult to identify the alphabet, as in the case of ancient scripts, this method is highly effective in identifying the alphabet. Furthermore, this technique performed well when it came to distinguishing Arabic text, including calligraphy. With the zoning system, the aim is to decrease the number of different element classes that are associated with the total number of alphabets used in Arabic cursive writing. Using the pixel value origin and center reign (CR) technique, which is coupled with letter morphology to accomplish complete word-level identification, the components are identified and the components are identified. Using the upper baseline and lower baseline together, this proposed technique produces a consistent Arabic pattern, which is intended to improve identification rates by increasing the number of matches. For Mediterranean keywords (cities in Algeria and Tunisia), the suggested approach makes use of a “Adapted Quadratic Discriminant Function Classifier-Based” shortcut, which is defined as follows: The results indicate that the correctness of the Othmani and Arabic scripts is more than 98.14 percent and 90.16 percent, respectively, based on 84 and 117 verses, respectively. As a consequence of the auditing method and the assessment section structure and software, the major problems were identified, with a few of them being specifically highlighted.
However, authors are recommended to consider the following concerns,
1. I can not see the paper in the journal template, but it seems in some other journal template
2. A thorough proofread is recommended
3. A few of the figures can be enhanced in terms of making them more readable
4. Author may elaborate clearly in the introduction section about their contribution
5. Authors need to clarify the scope of the work that their technique will work only for the Arabic context or for other related languages as well
6. In addition, authors may clearly mention about their scope that either their approach will work for Quranic Arabic, OR now a days practicing Arabic as well. As I can see a much difference among them
7. Better to update the conclusion section.
8. Better to update and add more latest and related references as well.
Authors may elaborate more on their this statement (The results indicate that the correctness of the Othmani and Arabic scripts is more than 98.14 percent and 90.16 percent, respectively, based on 84 and 117 verses, respectively. As a consequence of the auditing method and the assessment section structure and software, the major problems were identified, with a few of them being specifically highlighted. )

Experimental design

The authors presented a better work entitled, A deep learning for segmentation based on baseline approach for extracting Arabic Handwriting: Applied Calligraphy and Old Cursive.
The authors claims that According to the findings of this research, a novel technique for segmenting Arabic offline text is presented, in which the core splitter of the “Middle” portions from the “Lower” zone is identified by detecting character sharps degeneration in zones. With the exception of script localization and the essential feature of determining which direction a starting point is pointing, the baseline also functions as a delimiter for horizontal projections. Despite the fact that the bottom half of the characteristics is utilized to differentiate the modifiers in zones, the top half of the characteristics is not. Especially effective when the baseline is capable of separating characteristics into the bottom zone and the middle zone in a complex pattern where it is difficult to identify the alphabet, as in the case of ancient scripts, this method is highly effective in identifying the alphabet. Furthermore, this technique performed well when it came to distinguishing Arabic text, including calligraphy. With the zoning system, the aim is to decrease the number of different element classes that are associated with the total number of alphabets used in Arabic cursive writing. Using the pixel value origin and center reign (CR) technique, which is coupled with letter morphology to accomplish complete word-level identification, the components are identified and the components are identified. Using the upper baseline and lower baseline together, this proposed technique produces a consistent Arabic pattern, which is intended to improve identification rates by increasing the number of matches. For Mediterranean keywords (cities in Algeria and Tunisia), the suggested approach makes use of a “Adapted Quadratic Discriminant Function Classifier-Based” shortcut, which is defined as follows: The results indicate that the correctness of the Othmani and Arabic scripts is more than 98.14 percent and 90.16 percent, respectively, based on 84 and 117 verses, respectively. As a consequence of the auditing method and the assessment section structure and software, the major problems were identified, with a few of them being specifically highlighted.
However, authors are recommended to consider the following concerns,
1. I can not see the paper in the journal template, but it seems in some other journal template
2. A thorough proofread is recommended
3. A few of the figures can be enhanced in terms of making them more readable
4. Author may elaborate clearly in the introduction section about their contribution
5. Authors need to clarify the scope of the work that their technique will work only for the Arabic context or for other related languages as well
6. In addition, authors may clearly mention about their scope that either their approach will work for Quranic Arabic, OR now a days practicing Arabic as well. As I can see a much difference among them
7. Better to update the conclusion section.
8. Better to update and add more latest and related references as well.
Authors may elaborate more on their this statement (The results indicate that the correctness of the Othmani and Arabic scripts is more than 98.14 percent and 90.16 percent, respectively, based on 84 and 117 verses, respectively. As a consequence of the auditing method and the assessment section structure and software, the major problems were identified, with a few of them being specifically highlighted. )

Validity of the findings

The authors presented a better work entitled, A deep learning for segmentation based on baseline approach for extracting Arabic Handwriting: Applied Calligraphy and Old Cursive.
The authors claims that According to the findings of this research, a novel technique for segmenting Arabic offline text is presented, in which the core splitter of the “Middle” portions from the “Lower” zone is identified by detecting character sharps degeneration in zones. With the exception of script localization and the essential feature of determining which direction a starting point is pointing, the baseline also functions as a delimiter for horizontal projections. Despite the fact that the bottom half of the characteristics is utilized to differentiate the modifiers in zones, the top half of the characteristics is not. Especially effective when the baseline is capable of separating characteristics into the bottom zone and the middle zone in a complex pattern where it is difficult to identify the alphabet, as in the case of ancient scripts, this method is highly effective in identifying the alphabet. Furthermore, this technique performed well when it came to distinguishing Arabic text, including calligraphy. With the zoning system, the aim is to decrease the number of different element classes that are associated with the total number of alphabets used in Arabic cursive writing. Using the pixel value origin and center reign (CR) technique, which is coupled with letter morphology to accomplish complete word-level identification, the components are identified and the components are identified. Using the upper baseline and lower baseline together, this proposed technique produces a consistent Arabic pattern, which is intended to improve identification rates by increasing the number of matches. For Mediterranean keywords (cities in Algeria and Tunisia), the suggested approach makes use of a “Adapted Quadratic Discriminant Function Classifier-Based” shortcut, which is defined as follows: The results indicate that the correctness of the Othmani and Arabic scripts is more than 98.14 percent and 90.16 percent, respectively, based on 84 and 117 verses, respectively. As a consequence of the auditing method and the assessment section structure and software, the major problems were identified, with a few of them being specifically highlighted.
However, authors are recommended to consider the following concerns,
1. I can not see the paper in the journal template, but it seems in some other journal template
2. A thorough proofread is recommended
3. A few of the figures can be enhanced in terms of making them more readable
4. Author may elaborate clearly in the introduction section about their contribution
5. Authors need to clarify the scope of the work that their technique will work only for the Arabic context or for other related languages as well
6. In addition, authors may clearly mention about their scope that either their approach will work for Quranic Arabic, OR now a days practicing Arabic as well. As I can see a much difference among them
7. Better to update the conclusion section.
8. Better to update and add more latest and related references as well.
Authors may elaborate more on their this statement (The results indicate that the correctness of the Othmani and Arabic scripts is more than 98.14 percent and 90.16 percent, respectively, based on 84 and 117 verses, respectively. As a consequence of the auditing method and the assessment section structure and software, the major problems were identified, with a few of them being specifically highlighted. )

Additional comments

No additional comments, please refer to the previous set of comments.

Reviewer 2 ·

Basic reporting

The authors presented their work entitled A deep learning for segmentation based on baseline approach for extracting Arabic Handwriting: Applied Calligraphy and Old Cursive. Where authors claimed that, According to the findings of this research, a novel technique for segmenting Arabic offline text is presented, in which the core splitter of the “Middle” portions from the “Lower” zone is identified by detecting character sharps degeneration in zones. With the exception of script localization and the essential feature of determining which direction a starting point is pointing, the baseline also functions as a delimiter for horizontal projections. Despite the fact that the bottom half of the characteristics is utilized to differentiate the modifiers in zones, the top half of the characteristics is not. Especially effective when the baseline is capable of separating characteristics into the bottom zone and the middle zone in a complex pattern where it is difficult to identify the alphabet, as in the case of ancient scripts, this method is highly effective in identifying the alphabet. Furthermore, this technique performed well when it came to distinguishing Arabic text, including calligraphy. With the zoning system, the aim is to decrease the number of different element classes that are associated with the total number of alphabets used in Arabic cursive writing. Using the pixel value origin and center reign (CR) technique, which is coupled with letter morphology to accomplish complete word-level identification, the components are identified and the components are identified. Using the upper baseline and lower baseline together, this proposed technique produces a consistent Arabic pattern, which is intended to improve identification rates by increasing the number of matches. For Mediterranean keywords (cities in Algeria and Tunisia), the suggested approach makes use of a “Adapted Quadratic Discriminant Function Classifier Based” shortcut, which is defined as follows: The results indicate that the correctness of the Othmani and Arabic scripts is more than 98.14 percent and 90.16 percent, respectively, based on 84 and 117 verses, respectively. As a consequence of the auditing method and the assessment section structure and software, the major problems were identified, with a few of them being specifically highlighted.

However, authors are recommended to consider the following,
1. I can see the abstract is lengthier, which can be shorten and to the point.
2. Authors may elaborate on the novelty of the work in the introduction section.
3. Authors may elaborate a bit more on the research methodology
4. A few of the resolutions of the pictures can be increased
5. Authors may provide further details on their proposed approach, Proposed approaches for zone-based word recognition.
6. The conclusion can be revised and made it shorter for better understanding.
7. Overall, provided references are better enough. However, authors further can add the latest and related references.

Experimental design

The authors presented their work entitled A deep learning for segmentation based on baseline approach for extracting Arabic Handwriting: Applied Calligraphy and Old Cursive. Where authors claimed that, According to the findings of this research, a novel technique for segmenting Arabic offline text is presented, in which the core splitter of the “Middle” portions from the “Lower” zone is identified by detecting character sharps degeneration in zones. With the exception of script localization and the essential feature of determining which direction a starting point is pointing, the baseline also functions as a delimiter for horizontal projections. Despite the fact that the bottom half of the characteristics is utilized to differentiate the modifiers in zones, the top half of the characteristics is not. Especially effective when the baseline is capable of separating characteristics into the bottom zone and the middle zone in a complex pattern where it is difficult to identify the alphabet, as in the case of ancient scripts, this method is highly effective in identifying the alphabet. Furthermore, this technique performed well when it came to distinguishing Arabic text, including calligraphy. With the zoning system, the aim is to decrease the number of different element classes that are associated with the total number of alphabets used in Arabic cursive writing. Using the pixel value origin and center reign (CR) technique, which is coupled with letter morphology to accomplish complete word-level identification, the components are identified and the components are identified. Using the upper baseline and lower baseline together, this proposed technique produces a consistent Arabic pattern, which is intended to improve identification rates by increasing the number of matches. For Mediterranean keywords (cities in Algeria and Tunisia), the suggested approach makes use of a “Adapted Quadratic Discriminant Function Classifier Based” shortcut, which is defined as follows: The results indicate that the correctness of the Othmani and Arabic scripts is more than 98.14 percent and 90.16 percent, respectively, based on 84 and 117 verses, respectively. As a consequence of the auditing method and the assessment section structure and software, the major problems were identified, with a few of them being specifically highlighted.

However, authors are recommended to consider the following,
1. I can see the abstract is lengthier, which can be shorten and to the point.
2. Authors may elaborate on the novelty of the work in the introduction section.
3. Authors may elaborate a bit more on the research methodology
4. A few of the resolutions of the pictures can be increased
5. Authors may provide further details on their proposed approach, Proposed approaches for zone-based word recognition.
6. The conclusion can be revised and made it shorter for better understanding.
7. Overall, provided references are better enough. However, authors further can add the latest and related references.

Validity of the findings

The authors presented their work entitled A deep learning for segmentation based on baseline approach for extracting Arabic Handwriting: Applied Calligraphy and Old Cursive. Where authors claimed that, According to the findings of this research, a novel technique for segmenting Arabic offline text is presented, in which the core splitter of the “Middle” portions from the “Lower” zone is identified by detecting character sharps degeneration in zones. With the exception of script localization and the essential feature of determining which direction a starting point is pointing, the baseline also functions as a delimiter for horizontal projections. Despite the fact that the bottom half of the characteristics is utilized to differentiate the modifiers in zones, the top half of the characteristics is not. Especially effective when the baseline is capable of separating characteristics into the bottom zone and the middle zone in a complex pattern where it is difficult to identify the alphabet, as in the case of ancient scripts, this method is highly effective in identifying the alphabet. Furthermore, this technique performed well when it came to distinguishing Arabic text, including calligraphy. With the zoning system, the aim is to decrease the number of different element classes that are associated with the total number of alphabets used in Arabic cursive writing. Using the pixel value origin and center reign (CR) technique, which is coupled with letter morphology to accomplish complete word-level identification, the components are identified and the components are identified. Using the upper baseline and lower baseline together, this proposed technique produces a consistent Arabic pattern, which is intended to improve identification rates by increasing the number of matches. For Mediterranean keywords (cities in Algeria and Tunisia), the suggested approach makes use of a “Adapted Quadratic Discriminant Function Classifier Based” shortcut, which is defined as follows: The results indicate that the correctness of the Othmani and Arabic scripts is more than 98.14 percent and 90.16 percent, respectively, based on 84 and 117 verses, respectively. As a consequence of the auditing method and the assessment section structure and software, the major problems were identified, with a few of them being specifically highlighted.

However, authors are recommended to consider the following,
1. I can see the abstract is lengthier, which can be shorten and to the point.
2. Authors may elaborate on the novelty of the work in the introduction section.
3. Authors may elaborate a bit more on the research methodology
4. A few of the resolutions of the pictures can be increased
5. Authors may provide further details on their proposed approach, Proposed approaches for zone-based word recognition.
6. The conclusion can be revised and made it shorter for better understanding.
7. Overall, provided references are better enough. However, authors further can add the latest and related references.

Additional comments

The authors presented their work entitled A deep learning for segmentation based on baseline approach for extracting Arabic Handwriting: Applied Calligraphy and Old Cursive. Where authors claimed that, According to the findings of this research, a novel technique for segmenting Arabic offline text is presented, in which the core splitter of the “Middle” portions from the “Lower” zone is identified by detecting character sharps degeneration in zones. With the exception of script localization and the essential feature of determining which direction a starting point is pointing, the baseline also functions as a delimiter for horizontal projections. Despite the fact that the bottom half of the characteristics is utilized to differentiate the modifiers in zones, the top half of the characteristics is not. Especially effective when the baseline is capable of separating characteristics into the bottom zone and the middle zone in a complex pattern where it is difficult to identify the alphabet, as in the case of ancient scripts, this method is highly effective in identifying the alphabet. Furthermore, this technique performed well when it came to distinguishing Arabic text, including calligraphy. With the zoning system, the aim is to decrease the number of different element classes that are associated with the total number of alphabets used in Arabic cursive writing. Using the pixel value origin and center reign (CR) technique, which is coupled with letter morphology to accomplish complete word-level identification, the components are identified and the components are identified. Using the upper baseline and lower baseline together, this proposed technique produces a consistent Arabic pattern, which is intended to improve identification rates by increasing the number of matches. For Mediterranean keywords (cities in Algeria and Tunisia), the suggested approach makes use of a “Adapted Quadratic Discriminant Function Classifier Based” shortcut, which is defined as follows: The results indicate that the correctness of the Othmani and Arabic scripts is more than 98.14 percent and 90.16 percent, respectively, based on 84 and 117 verses, respectively. As a consequence of the auditing method and the assessment section structure and software, the major problems were identified, with a few of them being specifically highlighted.

However, authors are recommended to consider the following,
1. I can see the abstract is lengthier, which can be shorten and to the point.
2. Authors may elaborate on the novelty of the work in the introduction section.
3. Authors may elaborate a bit more on the research methodology
4. A few of the resolutions of the pictures can be increased
5. Authors may provide further details on their proposed approach, Proposed approaches for zone-based word recognition.
6. The conclusion can be revised and made it shorter for better understanding.
7. Overall, provided references are better enough. However, authors further can add the latest and related references.

·

Basic reporting

I can see the authors presented work is better and targeting with A deep learning approach for segmentation based on baseline approach for extracting Arabic Handwriting: Applied Calligraphy and Old Cursive
However, authors are recommended to consider the following,
1. Proofread is recommended to avoid the typos
2. The abstract can be revised/rephrased for better understanding
3. The novelty of the work is better enough. However, authors may pin point about their research contributions.
4. Authors may elaborate a bit more on the research methodology
5. I believe figures can be improved further
6. Authors may provide motivation, whey they chose the Arabic Language
7. The conclusion needs to be concise and clear

Experimental design

Authors may elaborate a bit more on the research methodology

Validity of the findings

The conclusion needs to be concise and clear

Additional comments

Overall, provided references are better enough. However, authors further can add the latest and related references.

---

## Round 0.2 · Minor Revisions

Some final comments to address

·

Basic reporting

Authors used deep learning model for segmenting arabia offline text

1- I have seen there is old reference in background study, therefore, I asked the author to add new recent reference about the deep learning in different domain


2- Authors should add research gap in introduction section

Experimental design

1- add confusion metrics to shows the researcher the performance of your model
2- the performance model and accuracy loss is not available
3 i have seen that there is research article has been achieved better than your model like
https://www.mdpi.com/2813-0324/2/1/14
4 I didn't see the structure of the deep learning model in your article
5. What are the add values in this article ?

Validity of the findings

The artcle need alot of improvemnt

Additional comments

No

---

## Round 0.3 · Minor Revisions

The authors have sincerely addressed all the issues raised by the reviewers. However, the title of the paper reads very awkward. I suggest changing it to: A deep learning based approach for extracting Arabic handwriting: applied calligraphy and old cursive.

---

## Round 0.4 · accepted · Accept

Your submission is ready to be accepted for publication.